# Respiratory cryptosporidiosis in Malawian children with diarrheal disease

**Pui-Ying Iroh Tam**[1,2,3]*, **Mphatso Chisala**[1], **Wongani Nyangulu**[1¤], **Herbert Thole**[1], **James Nyirenda**[1]

**1** Paediatrics and Child Health Research Group, Malawi-Liverpool Wellcome Trust Clinical Research Programme, Blantyre, Malawi, **2** Department of Paediatrics, University of Malawi College of Medicine, Blantyre, Malawi, **3** Department of Clinical Sciences, Liverpool School of Tropical Medicine, Liverpool, United Kingdom

¤ Current address: University of Malawi College of Medicine, Blantyre, Malawi
* irohtam@mlw.mw

**Editor:** jong-Yil Chai, Seoul National University College of Medicine, REPUBLIC OF KOREA

**Data Availability Statement:** Data cannot be shared publicly because of these data are owned by the Government of Malawi. Data are available from the University of Malawi College of Medicine

## Abstract

### Background

Respiratory cryptosporidiosis has been documented in children with diarrhea. We sought to describe the dynamics of respiratory involvement in children hospitalized with gastrointestinal (GI) diarrheal disease.

### Methods

We conducted a prospective, observational longitudinal study of Malawian children 2–24 months hospitalized with diarrhea. Nasopharyngeal (NP) swabs, induced sputum and stool specimens were collected. Participants that were positive by *Cryptosporidium* PCR in any of the three compartments were followed up with fortnightly visits up to 8 weeks post-enrollment.

### Results

Of the 162 children recruited, participants had mild-moderate malnutrition (mean HAZ -1.6 (SD 2.1)), 37 (21%) were PCR-positive for *Cryptosporidium* at enrollment (37 stool, 11 sputum, and 4 NP) and 27 completed the majority of follow-up visits (73%). *Cryptosporidium* was detected in all compartments over the 4 post-enrollment visits, most commonly in stool (100% at enrollment with mean cycle thresholds (Ct) of 28.8±4.3 to 44% at 8 weeks with Ct 29.9±4.1), followed by sputum (31% at enrollment with mean Ct 31.1±4.4 to 20% at 8 weeks with Ct 35.7±2.6), then NP (11% with mean Ct 33.5±1.0 to 8% with Ct 36.6±0.7). Participants with *Cryptosporidium* detection in both the respiratory and GI tract over the study period reported respiratory and GI symptoms in 81% and 62% of study visits, respectively, compared to 68% and 27%, respectively, for those with only GI detection, and had longer GI shedding (17.5±6.6 v. 15.9±2.9 days).

Research Ethics Committee for researchers who meet the criteria for access to confidential data (comrec@medcol.mw).

**Funding:** The work was supported by the Bill & Melinda Gates Foundation (OPP1191165 to PI). The funders had no role in study design, data collection and analysis, decision to publish, or preparation of the manuscript.

**Competing interests:** I have read the journal's policy and the authors of this manuscript have the following competing interests: PI has received grants from BMGF outside of the submitted work. All other authors have declared that no competing interests exist.

## Conclusion

*Cryptosporidium* was detected in both respiratory and GI tracts throughout the 8 weeks post-enrollment. The development of therapeutics for *Cryptosporidium* in children should target the respiratory as well as GI tract.

## Author summary

We conducted a prospective, observational longitudinal study of Malawian children 2–24 months hospitalized with diarrhea. NP swabs, induced sputum and stool specimens were collected. Participants that were positive by *Cryptosporidium* PCR in any of the three compartments were followed up with fortnightly visits up to 8 weeks post-enrollment. *Cryptosporidium* was detected by PCR in 21%, 7% and 3% in stool, sputum and nasopharynx of children hospitalized with diarrhea. Of those positive at enrollment, detection was noted in 44%, 20%, and 8%, respectively, by 8 weeks post-enrollment.

## Introduction

Cryptosporidiosis is a cause of diarrhea [1–3], excess mortality [4,5], stunting [1,6], and is associated with malnutrition [6,7]. The Global Enteric Multicenter Study (GEMS) identified *Cryptosporidium* as second most common cause of diarrhea among infants (0–11 months) in all four African countries studied (The Gambia, Mali, Mozambique, Kenya) regardless of HIV prevalence, and among the top five causes for older children (12–23 months) [4]. Respiratory cryptosporidiosis has been documented in up to a third of children presenting with diarrhea [8]; furthermore, respiratory detection without intestinal involvement has been reported [9,10], raising the possibility of primary respiratory infection with *Cryptosporidium*, either by inhalation or by contact with fomites [11].

Studies have identified *Cryptosporidium* in sputum but have not looked at whether respiratory involvement is a transient phenomenon or a reservoir for gastrointestinal (GI) disease. This has implications for therapeutic development. Our primary objective was to evaluate whether respiratory involvement of *Cryptosporidium* is a transient phenomenon in diarrheal disease, and to assess for respiratory and GI cryptosporidiosis concurrently and longitudinally in children hospitalized with diarrheal disease.

## Methods

### Ethics statement

The study was approved by the University of Malawi College of Medicine Research Ethics Committee (P.07/18/2438) and the Liverpool School of Tropical Medicine Research Ethics Committee (18–066). Written consent was obtained from parents or guardians prior to enrollment.

### Study design, setting, and participants

We conducted a prospective, observational-longitudinal study of Malawian children hospitalized with diarrhea [12]. Children aged 2–24 months and presenting with primary GI symptoms to Queen Elizabeth Central Hospital (QECH) in Blantyre, Malawi, were screened.

Eligible patients were those with at least three or more loose stools within the past 24 hours. Those with dysentery, or visible blood in loose stools, were excluded.

## Clinical procedures

Once written consent was obtained, a detailed history and physical exam was conducted and all then enrolled subjects had a NP swab, induced sputum, and stool sample collected. For induced sputum collection, subjects were given a nebulized 3% sodium chloride mist to inhale for 5–15 minutes. For infants, suctioning of the oropharynx was done after nebulization to collect the specimen. To ensure safety of the subject, induced sputum was only collected if there were no contraindications (based on PERCH criteria for induced sputum collection) [13]: severe hypoxia <92% on supplemental oxygen; inability to protect airways; severe bronchospasm at admission (defined as continued hypoxia <92% after appropriate bronchodilator therapy, with other markers of respiratory distress); seizure within the past 24 hours; or deemed inappropriate by the clinician for another reason (e.g. midface trauma, inhalational injury, pulmonary effusion, congestive heart failure, congenital heart disease, etc). If the above symptoms/conditions resolved within 48 hours, induced sputum collection was reconsidered at that point.

HIV rapid testing was conducted on infants and, if positive, infants were referred to the pediatric HIV clinic for further care. Any relevant lab investigations (including full blood count, urea and electrolytes, liver function tests, malaria parasite screen, blood and/or CSF culture, TB GeneXpert and AFB testing) performed as part of routine care were recorded.

All participating children had enrollment NP, sputum and stool specimens evaluated for *Cryptosporidium* by PCR. Sputum quality was evaluated by microscopy (good quality: ≤10 squamous epithelial cells/high powered field). Only subjects where *Cryptosporidium* was detected by PCR on any one of NP/sputum/stool were followed up in the post-enrollment phase up to 8 weeks post-enrollment, with a follow-up visit every two weeks for evaluation of symptoms, physical assessment, as well as NP/sputum/stool sampling.

## Laboratory procedures

DNA was extracted in stool using QIAamp Fast Stool Mini Kit (Qiagen, Hilden, Germany) and for respiratory tract using the QIAamp DNA mini kit (Qiagen) for *Cryptosporidium* [14], and UCP Mini kit (Qiagen) for respiratory co-pathogens. *Cryptosporidium* detection of specimens at enrollment were measured using quantitative polymerase chain reaction (qPCR), with appropriate positive and negative controls. In subsequent follow-up samples in stool, *Cryptosporidium* was detected using qPCR in a TaqMan Array Card (Thermo Fisher, Waltham, MA) using a custom design developed at the Houpt Laboratory (Charlottesville, VA) [15] and tested for 28 enteropathogens: rotavirus, norovirus GII, adenovirus, astrovirus, sapovirus, enterotoxigenic *Escherichia coli* (ETEC), enteropathogenic *E. coli* (EPEC), enteroaggregative *E. coli* (EAEC), Shiga-toxigenic *E. coli* (STEC), Shigella/enteroinvasive *E. coli* (EIEC), *Salmonella*, *Campylobacter jejuni/C. coli*, *Vibrio cholerae*, *Clostridium difficile*, *Cryptosporidium*, *Giardia lamblia*, *Entamoeba histolytica*, *Ascaris lumbricoides*, and *Trichuris trichiura*. In sputum, for multiplex PCR we used the respiratory pathogens 33 kit (Fast Track Diagnostics, Luxembourg, Luxembourg) which detected: *S. pneumoniae*, *S. aureus*, *M. catarrhalis*, *B. pertussis*, *H. influenzae* and *H. influenzae* type b, *C. pneumoniae*, *M. pneumoniae*, *K. pneumoniae*, *L. pneumophila*, and *Salmonella* species, influenza A/B/C, RSV A/B, parainfluenza virus types 1–4, coronaviruses NL63, 229E, OC43, and HKU1, human metapneumovirus A/B, rhinovirus, adenovirus, enterovirus, parechovirus, bocavirus, cytomegalovirus, and *P. jirovecii*.

### Statistical analysis

Frequencies and proportions of observed levels were reported for binary and categorical variables, with exact binomial 95% confidence intervals given where appropriate. Comparisons were performed using the Fisher's exact test for binary and categorical variables, t-test (two groups) or ANOVA (three or more groups) for approximately normally distributed variables and Wilcoxon rank sum (2 groups) or Kruskal-Wallis (3 or more groups) tests for variables with severely nonparametric distributions. Statistical analysis was performed using Stata software, version 16, and statistical significance was set at 0.05.

## Results

From 1 March 2019 to 3 April 2020, 162 children were recruited into the study. Two children did not submit specimens. Of the remaining 160 children, 37 (21%) were positive in any one of the three samples collected (37 stool (21%), 11 sputum (7%), and 4 NP (2%)) and were entered into follow-up. The study was discontinued early due to COVID-19, by which time 27 children had completed at least three of the four follow-up visits (73%).

Among the 162 children who were hospitalized with diarrhea, the mean age was 11.6±5.0 months, and 59% were male (Table 1). Mean height-for-age (HAZ), weight-for-age (WAZ), and weight-for-height (WHZ) z scores were -1.6±2.1, -1.0±1.3, and 0.1±1.6, respectively. Fifteen percent of mothers were HIV-infected, and five of those children tested positive by HIV rapid test (3% of total). The participants had a mean of 4.5±1.4 persons in a household, and the majority of children had access to a pit toilet/latrine (95%), piped drinking water (78%), and harbored a residential animal in the compound (66%). Among the 37 children who were *Cryptosporidium* PCR-positive, the mean age was 11.8±4.9 months, 65% were male and two (6%) were positive by HIV rapid test. No significant differences in characteristics between children who were *Cryptosporidium*-positive versus -negative were noted (S1 Table).

Induced sputum quality at enrollment was good in 82%, poor in 9% and unreadable in 9%, with similar proportions of good quality sputum among *Cryptosporidium*-positive and -negative participants (81% and 82%, respectively). Over the 8-week period, *Cryptosporidium* was detected in both respiratory and GI tracts (Table 2): most commonly in stool (100% at enrollment to 44% at 8 weeks), followed by sputum (31% at enrollment to 20% at 8 weeks), then NP (11% to 8%). Lowest Ct counts were generally noted in the stool (mean Ct 28.8±4.3 at enrollment to 29.9±4.1 at 8 weeks), followed by sputum (mean Ct 31.1± 4.4 at enrollment to 35.7 ±2.6 at 8 weeks), and NP ((mean Ct 33.5±1.0 to 36.6±0.7); Fig 1). When stratified by age, infants 2–11 months had lower stool Ct counts compared to those 12–24 months (mean Ct 27.5±4.4 v. 30.2±3.5), but this was not statistically significant (S2 Table). We detected *Cryptosporidium* in respiratory but not GI tract in 6 study visits. All NP-positive subjects were also sputum-positive, with the exception of one case at enrollment, a 15 month-old male with no respiratory symptoms in whom *Cryptosporidium* was detected in the NP (Ct 35.6) and stool (Ct 35.7), but not sputum. Detection in the respiratory tract did not always correlate with symptoms (Fig 2A–2H). For individual participants in whom *Cryptosporidium* was detected at enrollment, detection was noted most consistently in the GI tract throughout the 8 weeks. Respiratory co-pathogens were detected in 87/104 (84%) visits over the follow-up period.

Respiratory symptoms were noted in 72% of *Cryptosporidium*-positive participants at enrollment, the most common symptom being cough (69%), but also included runny nose (46%), crackles (17%) and wheeze (11%). For those entered into the 8-week follow-up period, participants with *Cryptosporidium* positivity in the respiratory tract had respiratory symptoms in 23/43 (53%) of visits.

**Table 1. Characteristics of children hospitalized with diarrhea at enrollment.**

| Characteristic | Children with diarrhea (n = 162)[a] | *Cryptosporidium* -positive (n = 37) | *Cryptosporidium* -negative (n = 123) | P value |
|---|---|---|---|---|
| **Demographics** | | | | |
| Age, months, mean (SD) | 11.6 (5.0) | 11.8 (4.9) | 11.5 (5.1) | 0.897 |
| 2–11 months | 90 (56%) | 24 (60%) | 65 (53%) | 0.565 |
| 12–24 months | 72 (44%) | 13 (40%) | 58 (47%) | |
| Male sex | 95 (59%) | 24 (65%) | 70 (57%) | 0.649 |
| Mother is employed (%) | 72 (44%) | 20 (54%) | 52 (42%) | 0.208 |
| Number of household members (SD) | 4.5 (1.4) | 4.2(1.4) | 4.5(1.4) | 0.204 |
| Children <5 years (SD) | 1.3 (0.5) | 1.2 (0.6) | 1.3 (0.5) | 0.533 |
| Residential animals in the compound (%) | 107 (66%) | 22 (59%) | 84 (68%) | 0.359 |
| Facility for disposing feces in households (%) | | | | |
| Pit toilet/latrine | 154 (95%) | 36 (97%) | 116 (94%) | 0.686 |
| Pour/flush toilet | 8 (5%) | 1 (3%) | 7 (6%) | |
| **Child health indicators** | | | | |
| Mother HIV status (%) | | | | |
| Positive | 25 (15%) | 8 (22%) | 17 (14%) | 0.391 |
| Unknown | 55 (34%) | 10 (29%) | 42 (34%) | |
| Child HIV rapid test (%) | | | | |
| Positive | 5 (3%) | 2 (6%) | 3 (2%) | 0.158 |
| Unknown/not done | 120 (74%) | 24 (63%) | 94 (76%) | |
| Z score (SD) | | | | |
| HAZ | -1.6 (2.1) | -1.4 (2.1) | -1.6 (2.1) | 0.552 |
| WAZ | -1.0 (1.3) | -1.0 (1.3) | -0.9 (1.3) | 0.831 |
| WHZ | 0.1 (1.6) | -0.1 (1.6) | 0.2 (2.3) | 0.396 |
| MUAC, cm (SD) | 13.7 (1.4) | 13.7 (1.3) | 13.8 (1.4) | 0.692 |
| Bipedal edema (%) | 5 (3%) | 1 (3%) | 4 (3%) | 1.000 |
| **Enrollment vitals and symptoms** | | | | |
| Temperature, ˚C (SD) | 36.7 (0.7) | 36.7 (0.6) | 36.8 (0.7) | 0.422 |
| Respiratory rate, breaths/min (SD) | 36 (6.1) | 36.4 (8.4) | 35.9 (5.3) | 0.902 |
| Oxygen saturations, % (SD) | 98.6 (1.4) | 98.6 (1.4) | 98.7 (1.4) | 0.719 |
| GI symptoms (%) | | | | |
| Vomiting | 70 (43%) | 18 (48%) | 50 (41%) | 0.847 |
| Abdominal pain/ tenderness | 65 (40%) | 13 (35%) | 50 (41%) | 0.330 |
| Poor feeding | 66 (40%) | 14 (38%) | 50 (41%) | 0.945 |
| Dehydration | 90 (56%) | 21 (57%) | 68 (55%) | 0.845 |
| Respiratory symptoms (%) | | | | |
| Cough | 98 (61%) | 25 (68%) | 71 (58%) | 0.283 |
| Runny nose | 58 (36%) | 17 (46%) | 39 (32%) | 0.150 |
| Difficulty in breathing | 3 (2%) | 1 (3%) | 2 (2%) | 0.531 |
| Wheezing | 14 (9%) | 4 (11%) | 9 (7%) | 0.487 |
| Chest indrawings/retractions | 2 (1%) | 0 | 2 (2%) | 1.000 |
| Crackles | 17 (11%) | 6 (17%) | 10 (8%) | 0.125 |
| *Cryptosporidium*-PCR positive | | | | |
| Stool (%) | 37 (21%) | 37 (100%) | 0 | <0.001 |
| Induced sputum (%) | 11 (7%) | 11 (31%) | 0 | <0.001 |
| NP (%) | 4 (2%) | 4 (11%) | 0 | <0.001 |
| Sputum quality (%) | | | | |

*(Continued)*

**Table 1.** (Continued)

| Characteristic | Children with diarrhea (n = 162)[a] | *Cryptosporidium* -positive (n = 37) | *Cryptosporidium* -negative (n = 123) | P value |
|---|---|---|---|---|
| Good (≤10 sq epis/hpf) | 131/160 (82%) | 30 (81%) | 101 (82%) | 0.013 |
| Poor (>10 sq epis/hpf) | 15/160 (9%) | 1 (3%) | 14 (12%) | |
| Unreadable[b] | 14/160 (9%) | 6 (17%) | 8 (7%) | |
| **Outcomes** | | | | |
| Mortality, inpatient (%) | NA | 0 | NA | |
| Mortality, post-discharge (%) | NA | 1 (3%) | NA | |

GI, gastrointestinal; HAZ, Height-for-age z score; HPF, high powered field; MUAC, mid-upper arm circumference; NA, not applicable; SD, standard deviation; WAZ, Weight-for-age z score; WHZ, Weight-for-height z score

[a]2/162 withdrew at enrolment, 1 submitted a respiratory but not a stool sample.

[b]Unable to assess sputum quality due to poor appearance.

Among participants with *Cryptosporidium* detection in both the respiratory and GI tract at enrollment compared to those with *Cryptosporidium* detection in the GI tract only (Table 3), a larger proportion reported respiratory symptoms (90% v. 65%), and GI shedding of *Cryptosporidium* was longer (14.3±2.1 v. 14.1±0.7 days), but these were not statistically significant. Among participants whom *Cryptosporidium* was detected in both the respiratory and GI tract at any point over the 8-week study period compared to GI tract only, a larger proportion reported respiratory (81% v. 68%) and GI symptoms (62% v. 27%) per study visit, and GI shedding of *Cryptosporidium* was longer (17.5±6.6 v. 15.9±2.9 days), but again these were not statistically significant.

## Discussion

This is the first longitudinal study to evaluate the respiratory cryptosporidiosis in pediatric diarrheal disease. In children hospitalized with diarrheal disease, *Cryptosporidium* was detected most frequently in stool, followed by sputum and NP. The most common respiratory symptom was cough. Longitudinally, we detected *Cryptosporidium* in both respiratory and GI tracts through 8 weeks post-enrollment, usually at lowest Ct counts in stool, followed by sputum and then NP. Longer GI shedding of *Cryptosporidium* was noted among those where we detected *Cryptosporidium* in both the respiratory and GI tracts, compared to GI detection only, although findings did not reach statistical significance.

Our *Cryptosporidium* detection rate is far higher than has been reported in diarrhea studies in sub-Saharan Africa, with rates of positivity ranging from 9% in Kenya to 14.7% in Mozambique [4,8]. Subclinical *Cryptosporidium* infection was noted to be high (6%) in Tanzania [16]. A previous study done at QECH found *Cryptosporidium* to be the third most common cause of diarrhea with a prevalence in stool of 28% among cases [17]. In our study, the positivity rate was 21%, and this would be 12% if we used the GEMS diarrheagenic cutoffs [18]. This thus supports the high prevalence of *Cryptosporidium* infection among young children hospitalized with diarrheal illness in sub-Saharan Africa.

Few studies, however, have evaluated the prevalence of respiratory cryptosporidiosis in pediatric diarrheal disease. A study in Uganda detected *Cryptosporidium* in 35.4% of induced sputum samples from children presenting with diarrhea who were *Cryptosporidium*-positive in the stool [8]. In our study, cough was noted in almost two-thirds of children presenting with diarrhea, and the frequency of respiratory signs and symptoms tended to be higher, but were not significantly different between those with *Cryptosporidium*-positive and -negative stool samples. In addition to sampling the lower respiratory tract with induced sputum, we

**Table 2. Longitudinal detection of *Cryptosporidium* in stool and respiratory tract and associated symptoms.**

| Characteristics | Study period | | | | |
|---|---|---|---|---|---|
| | Enrollment (n = 37) | 2 weeks[a] (n = 27) | 4 weeks[b] (n = 24) | 6 weeks[c] (n = 26) | 8 weeks[d] (n = 25) |
| ***Cryptosporidium* detection** | | | | | |
| Detection in NP | 4 (11%) | 5 (19%) | 2 (8%) | 1 (4%) | 2 (8%) |
| NP Ct values (SD) | 33.5 (1.0) | 33.7 (2.0) | 35.6 (0.6) | 35.7 | 36.6 (0.7) |
| Detection in sputum | 11 (31%) | 10 (37%) | 10 (42%) | 8 (31%) | 5 (20%) |
| Sputum Ct values (SD) | 31.1 (4.4) | 29.9 (3.6) | 33.0 (2.6) | 34.3 (2.9) | 35.7 (2.6) |
| Detection in GI tract | 37 (100%) | 22 (81%) | 16 (67%) | 11 (42%) | 11 (44%) |
| Stool Ct values (SD) | 28.8 (4.3) | 26.0 (4.5) | 30.3 (2.8) | 31.3 (3.5) | 29.9 (4.1) |
| Detection in respiratory tract only | 0 | 0 | 1 (4%) | 3 (8%) | 2 (8%) |
| Detection in GI tract only | 27 (73%) | 12 (43%) | 7 (29%) | 5 (19%) | 8 (32%) |
| Detection in respiratory and GI tract | 10 (27%) | 9 (33%) | 9 (26%) | 6 (23%) | 3 (12%) |
| **Presentation** | | | | | |
| Referral to health facility for an illness in the past 7 days | 36 (100%) | 10 (37%) | 6 (25%) | 5 (19%) | 7 (28%) |
| Diarrhea in the past 7 days | 36 (100%) | 0 | 2 (8%) | 4 (15%) | 3 (12%) |
| Respiratory symptoms in the past 7 days[e] | 26 (72%) | 9/10 (90%) | 4/6 (67%) | 2/5 (40%) | 5/7 (71%) |
| Cough | 24 (69%) | 5/10 (50%) | 2/6 (33%) | 2/5 (40%) | 5/7 (71%) |
| Runny nose | 16 (46%) | 7/10 (70%) | 4/6 (67%) | 1/5 (20%) | 3/7 (43%) |
| Difficulty in breathing | 1 (3%) | 0 | 0 | 0 | 0 |
| Wheezing | 4 (11%) | 3/10 (30%) | 1/6 (17%) | 1/5 (20%) | 0 |
| Chest indrawings/ retractions | 0 | 0 | 0 | 0 | 0 |
| Crackles | 6 (17%) | 5/10 (50%) | 1/6 (17%) | 1/5 (20%) | 0/7 |
| Vomiting | 16 (46%) | 0/10 | 0/6 | 1/5 (20%) | 0/7 |
| Abdominal pain/ tenderness | 11 (31%) | 1/10 (10%) | 1/6 (17%) | 1/5 (20%) | 0/7 |
| Poor feeding | 14 (40%) | 0 | 0 | 0 | 0 |
| Other household members with respiratory symptoms | 13 (37%) | 1/6 (17%) | 3/5 (60%) | 0 | 0 |
| Other household members with GI symptoms | 4 (11%) | 0 | 3 (13%) | 1 (4%) | 2 (8%) |
| **Vital signs** | | | | | |
| Respiratory rate, breaths/min (SD) | 36 (6.1) | 34.7 (3.4) | 32.8 (3.2) | 34.4 (5.7) | 32.3 (3.7) |
| Oxygen saturations, % (SD) | 98.6 (1.4) | 99.1 (1.2) | 98.5 (1.1) | 99.4 (0.8) | 98.8 (1.4) |
| **Outcome** | | | | | |
| Mortality | 1[f] (3%) | 0 | 0 | 0 | 0 |

Ct, cycle threshold; GI, gastrointestinal; NP, nasopharynx; SD, standard deviation

[a]Two children withdrew at enrollment, one child withdrew after enrollment, five did not enter follow-up due to early stopping of study, and three were lost to follow-up.

[b]Three children missed week 4 visit.

[c]One child missed week 6 visit.

[d]Two children did not complete week 8 visit due to early stopping of study.

[e]In follow-up study visits, symptoms were recorded only if these led to health facility referral.

[f]Patient died after discharge before the week 2 visit.

also collected NP specimens throughout the follow-up period, and documented detection throughout the 8-week period, although in fewer numbers, more intermittently, and at higher Ct counts compared to sputum. Respiratory symptoms were more common among those *Cryptosporidium*-positive in respiratory and GI tract compared to positive in GI tract alone, although again these findings were not statistically significant. In contrast to the Uganda study, which only collected sputum samples from children with cough, unexplained tachypnea or hypoxia, we collected respiratory specimens in all participants at enrollment and noted that

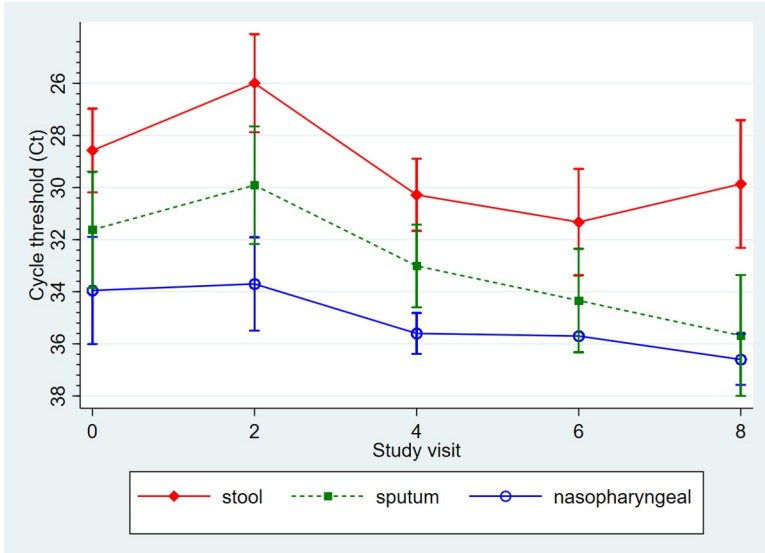

**Fig 1. Comparison of *Cryptosporidium* detection in GI and respiratory tracts (mean cycle threshold with 95% confidence intervals) over 8 weeks in children hospitalized with diarrheal disease.**

participants with *Cryptosporidium* positivity in the respiratory tract had respiratory symptoms in just over half of visits. These data support that almost half the participants with respiratory *Cryptosporidium* detection are asymptomatic, and is a lower, rather than upper, respiratory tract pathogen or colonizer [19–22].

The continued detection of *Cryptosporidium* in both GI and respiratory tract over 8 weeks may reflect the young age of this population. A study in Bangladesh noted that children ≤2 years positive for *Cryptosporidium* shed it for a mean of 4.1 weeks, which was significantly

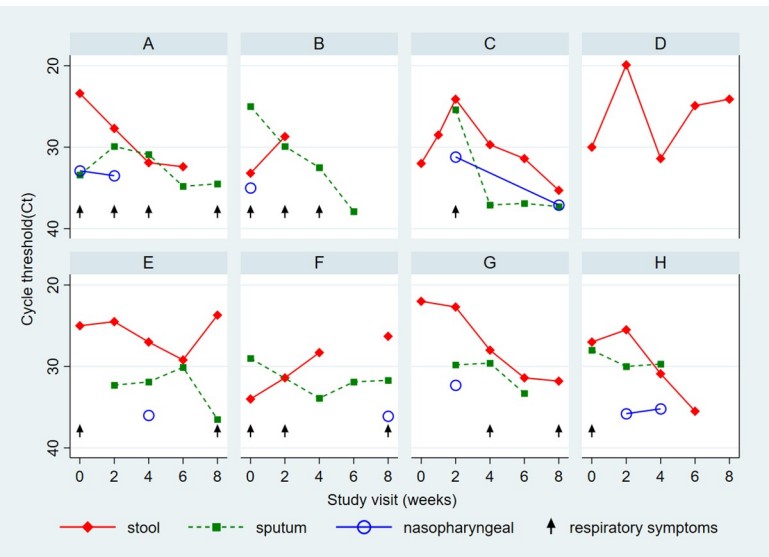

**Fig 2.** Patient-level dynamics of *Cryptosporidium* detection in GI and respiratory tracts over 8 weeks in a: A) 10 month-old; B) 14 month-old; C) 14 month-old; D) 7 month-old; E) 11 month-old; F) 11 month-old; G) 10 month-old; and H) 15 month-old.

**Table 3. Characteristics and associated symptoms in participants with *Cryptosporidium* detection in GI tract only v. GI and respiratory tract A) at enrollment; B) at any point throughout study period.**

| Characteristic | At enrollment (n = 37) | | | At any point throughout study period (n = 27) | | |
|---|---|---|---|---|---|---|
| | GI detection only (n = 27) | GI and respiratory detection (n = 10) | P value | GI detection only (n = 16) | GI and respiratory detection (n = 11) | P value |
| Age, months (SD) | 11.3 (5.3) | 13 (4.0) | 0.370 | 12.2 (2.3) | 12.0 (6.7) | 0.922 |
| Male sex | 18 (67%) | 6 (60%) | 0.7664 | 9 (56%) | 8 (72%) | 0.448 |
| Z score (SD) | | | | | | |
| HAZ | -1.4 (2.4) | -1.3 (1.1) | 0.896 | -1.6 (2.5) | -1.4 (1.9) | 0.837 |
| WAZ | -1.1 (1.4) | -0.6 (1.2) | 0.319 | -1.1 (1.2) | -0.8 (1.2) | 0.474 |
| WHZ | -0.2 (1.6) | 0.0 (1.7) | 0.708 | -0.1 (1.7) | 0.0 (1.6) | 0.945 |
| Number of household members (SD) | 4.2 (1.4) | 4.3 (1.5) | 0.771 | 4.4 (1.5) | 4.1 (0.9) | 0.504 |
| Household children <5 years | 1.3 (0.7) | 1.1 (0.3) | 0.449 | 1.1 (0.1) | 1.2 (0.1) | 0.696 |
| Pit latrine | 26 (96%) | 10 (100%) | 0.529 | 16 (100%) | 10 (90%) | 0.407 |
| Piped water for drinking | 4 (15%) | 1 (10%) | 0.916 | 2 (13%) | 3 (27%) | 0.370 |
| Piped water for utensils | 19 (69%) | 6 (60%) | 0.448 | 13 (81%) | 4 (45%) | 0.124 |
| Treated water (%) | 14 (50%) | 2 (20%) | 0.142 | 6 (37%) | 5 (45%) | 0.710 |
| Residential animals in the compound | 15 (54%) | 7 (70%) | 0.467 | 11 (69%) | 7 (63%) | 1.000 |
| Respiratory symptoms | 17 (65%) | 9 (90%) | 0.223 | 11 (68%) | 9 (81%) | 0.385 |
| Enrollment oxygen sats, % (SD) | 98.5 (1.4) | 98.8 (1.2) | 0.561 | 98.5% (1.1) | 98.3% (1.7) | 0.653 |
| GI symptoms | 24 (89%) | 8 (80%) | 0.603 | 3 (27%) | 10 (62%) | 0.120 |
| + | 14.1 (0.7) | 14.3 (2.1) | 0.819 | 15.9 (2.9) | 17.5 (6.6) | 0.445 |

GI, gastrointestinal; HAZ, Height-for-age z score; SD, standard deviation; WAZ, Weight-for-age z score; WHZ, Weight-for-height z score

longer than those >2 years (mean 1.7 weeks) [23]. However, that was a household transmission study and did not evaluate children hospitalized with diarrhea. For respiratory pathogens, prolonged detection of respiratory viruses has been noted in a community surveillance study in Utah, with significantly longer detection noted in children <5 years (mean 1.9 weeks) compared to other age groups (mean 1.6–1.7 weeks) [24]. To our knowledge, prolonged detection in the respiratory tract has not before been described for a protozoan parasite; this was not associated with higher prevalence of respiratory symptoms.

Prevalence of cryptosporidiosis is higher in cases of persistent v. acute diarrhea (15% v. 6.1% in a study in Guinea Bissau [5]). Persistent diarrhea has been described for cryptosporidiosis with duration of up to 5 months among severely immunocompromised infants [25]. In the Etiology, Risk Factors and Interactions of Enteric Infections and Malnutrition and the Consequences for Child Health and Development Program (MAL-ED) study, cryptosporidiosis was the fifth highest attributable pathogen in all pediatric diarrhea in the community setting, and had increased frequency among those with prolonged and severe diarrhea [3]. Among our participants, GI symptoms resolved within 2 weeks of hospitalization for all participants, but *Cryptosporidium* was detected in stool for eight participants throughout the 8-week period. Shedding duration for cryptosporidiosis among children hospitalized with diarrhea in sub-Saharan Africa has not been described previously using molecular techniques, although in Malawian HIV-infected adults we documented consistent shedding for up to 8 weeks [26]. Nutritional status may also account for the prolonged detection of up to 8 weeks that we documented in our study population. Even though the mean WHZ was within normal, the mean

HAZ and WAZ of recruited participants both met WHO criteria for stunting and wasting, and 41%, 22%, and 10% had HAZ, WAZ, and WHZ <-2, respectively, indicating mild-moderate malnutrition in this population. Reduced nutritional status could impact host immunity and predisposition to colonization/infection [27–29].

Cryptosporidiosis has been associated with excess mortality in children who had the infection in infancy, and this excess mortality persists into the second year of life [4,5]. In our study we had one death, which occurred in a 7 month-old breastfeeding, HIV unknown male. This child had anthropometric parameters consistent with severe acute malnutrition (mid-upper arm circumference 11 cm, HAZ -1.85, WAZ -3.01, WHZ -2.47, no nutritional edema) at time of enrollment, and an enrollment *Cryptosporidium* Ct of 23 in the stool but negative in sputum and NP. At time of enrollment he was afebrile with normal vitals and had mild dehydration, diarrhea associated with abdominal tenderness and vomiting, as well as runny nose, cough and wheeze, but no shortness of breath. Death occurred two weeks after hospitalization after the child developed a cough and fever and was treated at an outpatient health facility.

This study has several limitations. The sample size was small, and the study population was drawn from a single hospital, and therefore findings may not be generalizable to other settings. Secondly, we did not evaluate for respiratory co-pathogens in our enrollment specimens, thereby limiting our ability ascribe symptoms to a single pathogen, although we did document co-pathogen detection in 84% during the follow-up period. Thirdly, we did not genotype *Cryptosporidium* to ascertain whether re-infections occurred. However, these limitations are balanced out by the longitudinal observational design, which allowed us to document for the first time the dynamics of respiratory *Cryptosporidium* among children hospitalized with diarrheal disease.

## Conclusion

In summary, in this study we demonstrated that over 20% of young children hospitalized with diarrheal disease in our setting are positive for *Cryptosporidium*, and they can shed *Cryptosporidium* in the stool for up to 2 months. Concurrent respiratory cryptosporidiosis can be detected in a substantial proportion of young children, and detection continues over a prolonged period, with detection more prominent in the lower rather than upper respiratory tract. However, the small sample size limits our ability to make a definitive conclusion.

Nevertheless, this has implications for the development of therapeutics for cryptosporidiosis, which is limited currently to nitazoxanide, and which is poorly efficacious in malnourished children [30,31]. Since malnourished children are in greatest need for *Cryptosporidium* therapeutics, scientists have been actively trying to find new, more effective drugs to treat cryptosporidiosis in this population. Because safety is of paramount concern in this age group, those developing drugs have debated whether a drug that only resides in the GI tract would be adequate for treating cryptosporidiosis in malnourished children, compared to one with broader systemic distribution [32]. Our findings suggest that the respiratory tract is a significant reservoir for *Cryptosporidium* infection, and that to cure children, may require a drug that distributes to the lungs as well as the GI tract. Thus, future research on therapeutic development should focus on drugs that target not only the GI but also the respiratory tract.

## Supporting information

**S1 Table. Full characteristics of study population at enrollment.**
(DOCX)

**S2 Table.** *Cryptosporidium*-**positive at enrollment study population characteristics, stratified by age.**
(DOCX)

## Acknowledgments

We thank the patients and parents for participating in this study; research clinical and laboratory staff for conducting the study; David Moore and Tanja Adams with training the CryptoResp clinical and laboratory teams; Eric Houpt and Darwin Operario with providing TAC cards and technical assistance; Neema Toto for interim study support; Thokozani Ganiza for assistance with data management; and Wes Van Voorhis for reviewing drafts, and providing feedback and support throughout the project.

## Author Contributions

**Conceptualization:** Pui-Ying Iroh Tam.

**Data curation:** Mphatso Chisala, Wongani Nyangulu, James Nyirenda.

**Formal analysis:** Mphatso Chisala, James Nyirenda.

**Investigation:** Mphatso Chisala, Wongani Nyangulu, Herbert Thole, James Nyirenda.

**Methodology:** Pui-Ying Iroh Tam.

**Project administration:** Mphatso Chisala, Wongani Nyangulu.

**Resources:** Pui-Ying Iroh Tam.

**Supervision:** Pui-Ying Iroh Tam, Mphatso Chisala, Wongani Nyangulu.

**Validation:** James Nyirenda.

**Visualization:** Pui-Ying Iroh Tam.

**Writing – original draft:** Pui-Ying Iroh Tam.

**Writing – review & editing:** Pui-Ying Iroh Tam, Mphatso Chisala, Wongani Nyangulu, Herbert Thole, James Nyirenda.

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
