## [Decision Letter · Decision Letter 0]

18 Jun 2021

Dear Dr Iroh Tam,

Thank you very much for submitting your manuscript "Respiratory cryptosporidiosis in Malawian children with diarrheal disease" for consideration at PLOS Neglected Tropical Diseases. As with all papers reviewed by the journal, your manuscript was reviewed by members of the editorial board and by several independent reviewers. In light of the reviews (below this email), we would like to invite the resubmission of a significantly-revised version that takes into account the reviewers' comments. 

Your manuscript has been reviewed by 2 independent reviewers. Both of them suggested some revisions. Please see their comments below and in the attached files.

We cannot make any decision about publication until we have seen the revised manuscript and your response to the reviewers' comments. Your revised manuscript is also likely to be sent to reviewers for further evaluation.

Sincerely,

Jong-Yil Chai

Associate Editor

Pikka Jokelainen

Deputy Editor

Your manuscript has been reviewed by 2 independent reviewers. Both of them suggested some revisions. Please see their comments below and in the attached files.

Reviewer's Responses to Questions

**Key Review Criteria Required for Acceptance?**

**Methods**

-Are the objectives of the study clearly articulated with a clear testable hypothesis stated?

-Is the study design appropriate to address the stated objectives?

-Is the population clearly described and appropriate for the hypothesis being tested?

-Is the sample size sufficient to ensure adequate power to address the hypothesis being tested?

-Were correct statistical analysis used to support conclusions?

-Are there concerns about ethical or regulatory requirements being met?

Reviewer #1: See attachment.

Reviewer #2: The objectives of the study were clearly explained and a clear testable hypothesis was described. The study design was appropriate for the proposed hypothesis.

The study population was Malawian children aged 2-24 months and presenting with primary GI symptoms in Queen Elizabeth Central Hospital in Blantyre. This population was clearly described and appropriate for the proposed study. 

The sample of children who were positive and followed for the 8 weeks was too small to achieve some of the study objectives. 

The statisitical test could be have been developed more. 

There were no concerns about ethical considerations.

**Results**

-Does the analysis presented match the analysis plan?

-Are the results clearly and completely presented?

-Are the figures (Tables, Images) of sufficient quality for clarity?

Reviewer #1: See attachment.

Reviewer #2: The analysis is consistent with the analysis plan and the results are clearly presented.

The results presented in some of the tables and figures could be presented more concisely. Table 1 provides a lot of information about the characteristics of the enrolled children. This could be reduced especially since no significant differences were found between cryptosporidium infected children and non-infected children.

**Conclusions**

-Are the conclusions supported by the data presented?

-Are the limitations of analysis clearly described?

-Do the authors discuss how these data can be helpful to advance our understanding of the topic under study?

-Is public health relevance addressed?

Reviewer #1: See attachment.

Reviewer #2: The authors conclusions are partially supported by the results presented.

The limitations regarding the small sample size are not adequately addressed. 

The study is innovative and so does increase understanding of results concerning the dynamics of respiratory and GI cryptosporidium infections 

The public health relevance is addressed by the authors who suggest that therapeutic drug development may require a drug that targets the lungs as well as the GI tract to cure children.

**Editorial and Data Presentation Modifications?**

Reviewer #1: See attachment.

Reviewer #2: The results presented in some of the tables and figures could be presented more concisely.

**Summary and General Comments**

Reviewer #1: See attachment.

Reviewer #2: Overall the study in well-executed. The results concerning the dynamics of respiratory and GI cryptosporidium infections are interesting and important. The small sample size makes it difficult to arrive at definitive conclusion and so the authors should stress that in their conclusions.

I have suggested that the results presented in Table 1 could be stratified by age and or gender to have a better understanding of these dynamics.

PLOS authors have the option to publish the peer review history of their article (what does this mean?). If published, this will include your full peer review and any attached files.

Reviewer #1: No

Reviewer #2: No
---

## [Editor Report · Decision Letter 1]

11 Jul 2021

Dear Dr Iroh Tam,

We are pleased to inform you that your manuscript 'Respiratory cryptosporidiosis in Malawian children with diarrheal disease' has been provisionally accepted for publication in PLOS Neglected Tropical Diseases.

Regarding corrections to spelling, please carefully check that '*Cryptosporidium*' is spelled with capital first letter everywhere in the text. 

Best regards,

Jong-Yil Chai

Associate Editor

Pikka Jokelainen

Deputy Editor

Your revised manuscript has been reviewed by 3 independent reviewers. They gave no further comments. Thus, I am pleased to decide to accept your revised manuscript. A few things to revise (for example, all 'cryptosporidium' within the text should be 'Cryptosporidium') can be addressed while constructing the PDF.

---

## [Editor Report · Acceptance letter]

26 Jul 2021

Dear Dr Iroh Tam,

We are delighted to inform you that your manuscript, "Respiratory cryptosporidiosis in Malawian children with diarrheal disease," has been formally accepted for publication in PLOS Neglected Tropical Diseases.

Best regards,

Shaden Kamhawi

co-Editor-in-Chief

Paul Brindley

co-Editor-in-Chief
